# GRAWALKER: VULNERABILITY DETECTION VIA CODE SEMANTIC FUSION GRAPH WITH EDGE-AWARE RANDOM WALK

## ABSTRACT

Vulnerability detection plays a crucial role in software security. However, existing deep learning methods still face challenges in effectively capturing and learning both semantic and structural information from source code. In this paper, we propose a novel framework, **GraWalkER**, designed to achieve more accurate code vulnerability detection. GraWalkER employs a two-stage approach for vulnerability detection. In the first stage, we introduce a novel **Code Semantic Fusion Graph (CSFG)** to generate an enhanced code graph representation. This new graph structure incorporates multifaceted program properties from source code. In the second stage, we propose an **Edge-aware Random Walk with Unifying Memory (ERUM)** method. ERUM extracts path features by weighting edge types during random walk processes, enabling comprehensive graph representation learning for precise vulnerability identification. We conduct comprehensive evaluations of GraWalkER on three datasets covering both Java and C/CPP languages, comparing it against seven state-of-the-art vulnerability detection methods. Experimental results consistently demonstrate that GraWalkER outperforms all baseline methods in vulnerability detection tasks, validating its effectiveness.

## 1 INTRODUCTION

Vulnerability detection is a key technology for ensuring software quality and security Zhang et al. (2023); Chakraborty et al. (2021); Qiu et al. (2024). The increasing scale and complexity of software systems Wang et al. (2021) make it challenging for manual auditing and unit testing to cover all potential risks. Compounded by the diversity of programming languages and intricate interwoven control flows, this task presents significant challenges.

Advances in deep learning have propelled the application of neural networks to code defect detection Wu et al. (2022); Qiu et al. (2024). Some DL approaches linearize source code into token sequences for processing Scandariato et al. (2014), while others Wu et al. (2022); Han et al. (2022) employ Graph Neural Networks (GNNs) for code graph feature learning. Additionally, pre-trained code models Bi et al. (2025); Feng et al. (2020); Guo et al. (2020) are often applied to code detection tasks.However, these methods suffer from two main limitations:

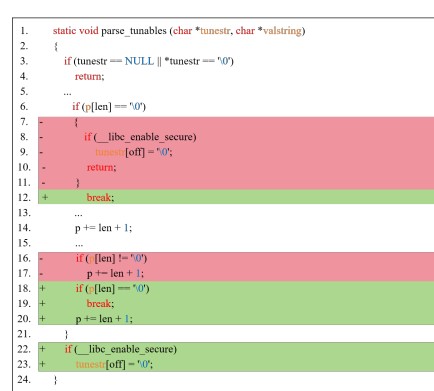

Figure 1: Case study: unchecked input length vulnerability in glibc's GLIBC_TUNABLES parser (CVE-2023-4911)

(1) **Incomplete Code Representation**: Current approaches fail to comprehensively capture both semantic and structural information. Sequence-based methods Li et al. (2021c); Russell et al. (2018) and graph-structure methods Nguyen et al. (2022); Wang et al. (2020a;b) capture only partial information, leading to performance degradation on complex code. (2) **Limitations of GNNs**: GNN-

based methods face issues of limited expressive power, over-smoothing, and information bottle-necks. Xu et al. (2018) demonstrated that GNNs cannot exceed the discriminative power of the WL test Weisfeiler & Leman (1968), and deep convolutional layers often lead to feature degradation Corso et al. (2020). Modeling critical paths (e.g., loop structures or cross-function dependencies) remains difficult. As shown in Fig. 1, some models failed to correctly interpret the semantics of __libc_enable_secure in lines 7-11, resulting in failure in vulnerability detection.

To address these limitations, this paper proposes **GraWalkER**, a novel framework integrating a multidimensional Code Semantic Fusion Graph (CSFG) with Edge-aware Random Walk with Uni-fying Memory (ERUM) for code vulnerability detection. The framework first constructs a novel multidimensional CSFG representation by combining key code semantic and structural information. Then, inspired by Wang & Cho (2024), GraWalkER introduces an ERUM method. ERUM generates stochastic walks for nodes, incorporating edge-type awareness during traversal to thoroughly char-acterize their topological context. Finally, a MLP classifier utilizes the aggregated representation of the entire CSFG graph for vulnerability prediction.

To evaluate the effectiveness of GraWalkER, we conducted comprehensive experiments on mul-tiple high-quality vulnerability datasets for Java and C/CPP languages. Results demonstrate that GraWalkER surpasses existing vulnerability detection methods, achieving superior performance.

To the best of our knowledge, this paper makes the following key contributions:

- We propose the **Code Semantic Fusion Graph (CSFG)** as a novel graph-structured code representation, which integrates Abstract Syntax Trees (ASTs), Control Flow Graphs (CFGs), Control Dependence Graphs (CDGs), Call Graphs (CGs), and Code Natural Se-quences (CNSs) to capture a more comprehensive view of code semantics and structure.

- We introduce the **Edge-aware Random Walk with Unifying Memory (ERUM)** method as a core component of GraWalkER, designed to replace traditional GNNs for feature learn-ing on code graphs, thereby achieving enhanced vulnerability detection performance.

- We conduct extensive experiments to evaluate the GraWalkER model. The results demon-strate that GraWalkER outperforms existing state-of-the-art baseline models, validating its effectiveness in vulnerability detection.

- We release our replication package[1], including source code and datasets, to facilitate repro-ducibility and future research.

## 2 RELATED WORK

Existing neural network detection methods can be primarily categorized into two classes: sequence-based modeling approaches and graph structure-based modeling approaches Li et al. (2021a). Sequence-based methods treat code as token sequences Li et al. (2021b;c). Scandariato et al. (2014) employed bag-of-words models for encoding. Additionally, some studies adopted hybrid RNN/CNN models Seas et al. (2024) to enhance modeling of latent defect features in serialized code. Graph structure-based methods utilize GNNs Kipf & Welling (2016) to model syntactic relationships Cao et al. (2022). REVEAL Chakraborty et al. (2021) which employed GNNs with resampling strategies to mitigate class imbalance; iGnnVD Chen et al. (2024) which fused multiple GNNs to model hy-brid property graphs, achieving improved detection performance. Despite progress, these traditional graph models predominantly assume homogeneous graph structures, failing to model semantic dif-ferences between edge types. This hinders their ability to capture multidimensional interactions in source code Wang & Cho (2024).

In DL-based code analysis tasks, the Code Property Graph (CPG) is a common graph representa-tion for code analysis, primarily including AST, CFG, and Program Dependence Graph (PDG). AST more closely reflects the semantic organization of programs Zhang & Saber (2025). Lin et al. (2018) achieved cross-project detection by serializing ASTs; Zhang et al. (2019) proposed AST de-composition (ASTNN) to capture syntactic knowledge from large ASTs. CFG is mainly used to cap-ture intra-procedural control flow characteristics. Tufano et al. (2018) learned structural similarities based on CFGs; Tang et al. (2023) represented function-level CFGs with CSGVD. PDG also plays a

---

[1]https://anonymous.4open.science/r/GraWalkER

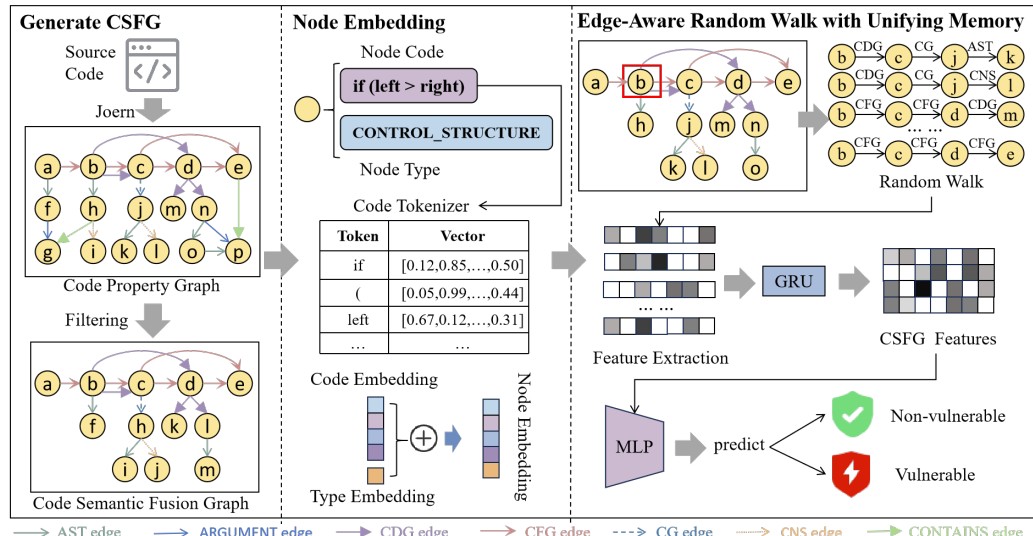

Figure 2: Overview of the GraWalkER Framework

critical role in capturing control and data dependencies. VulCNN Wu et al. (2022) utilized PDG and node centrality for vulnerability detection; VulChecker Mirsky et al. (2023) enhanced PDG at the instruction level to improve localization accuracy. While each representation has distinct advantages, they all suffer from structural incompleteness or underutilization of features. Efficiently modeling edge semantics and node contexts in heterogeneous structures remains a significant challenge yet to be addressed by current research approaches.

## 3 METHOD

In this section, we introduce our vulnerability detection framework GraWalkER. The overall workflow of our method is illustrated in Fig. 2.

### 3.1 PROBLEM DEFINITION

We consider vulnerability detection at the function level for source code, i.e., our goal is to identify whether a given function in the original source code is vulnerable. In this work, we define the data sample as $\{(c_i, y_i) | c_i \in \mathbb{C}, y_i \in \mathbb{Y}\}_{i=1}^n$, where $\mathbb{C}$ denotes the raw code dataset, $\mathbb{Y} = \{0, 1\}$ represents the corresponding label set (where 1 indicates the vulnerable label set and 0 otherwise), and $n$ is the number of instances.

In our work, we formulate the vulnerability detection task as a graph classification problem and address it using random walk-based graph feature extraction methods. Therefore, we first construct a corresponding CSFG $g_i(\mathcal{V}, \boldsymbol{X}, \boldsymbol{E}) \in \mathcal{G}$ for each source code $c_i$, where $\mathcal{V} = \{v_1, \ldots, v_m\}$ is the set of $m$ nodes in the graph; for any node $v_i \in \mathcal{V}$, it contains two attributes: the type attribute $vl(v_i)$ and the source code attribute $vc(v_i)$; $\boldsymbol{X} \in \mathbb{R}^{m \times d}$ denotes the node feature matrix, with each node $v_j \in \mathcal{V}$ represented by a $d$-dimensional vector $\boldsymbol{x}_j \in \mathbb{R}^d$; $\boldsymbol{E} = \{e_1, \ldots, e_v\}$ is the set of $v$ edges in the graph, and $e_i = (v_s, v_t, \tau)$ indicates that there exists an edge of type $\tau$ between the source node $v_s$ and the target node $v_t$, where $\tau \in \mathcal{T}$ represents the edge type attribute.

### 3.2 FEATURE EXTRACTION

To more comprehensively and accurately characterize the semantic information of source code, we propose a novel graph-level code representation method: the CSFG. It integrates semantic and structural information including the AST, CDG, CFG, CG and CNS. This approach not only encompasses the internal control flow and data flow information of source code functions, but also preserves both the natural sequence and logical order of the code.

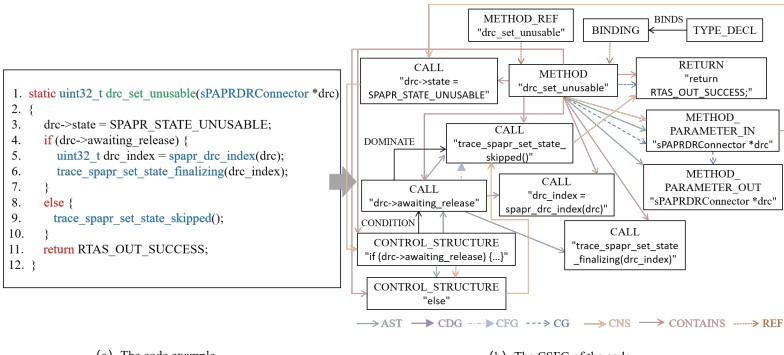

(a) The code example          (b) The CSFG of the code

Figure 3: An example of generating CSFG

### 3.2.1 GENERATING CODE SEMANTIC FUSION GRAPH

Next, we briefly introduce each code representation and explain how they are integrated into the CSFG.

- **Abstract Syntax Tree (AST)**: Tree representation of source code syntax. Nodes denote syntactic units (e.g., expressions, statements), with hierarchical relationships reflecting nesting structures.
- **Control Flow Graph (CFG)**: Models program execution paths. Nodes represent basic blocks, directed edges indicate control transfers (branches/loops).
- **Control Dependence Graph (CDG)**: Captures control dependencies between statements. Nodes link code blocks where execution depends on conditional evaluations.
- **Call Graph (CG)**: Maps function call hierarchies. Edges connect caller-callee relationships across the codebase.
- **Code Natural Sequence (CNS)**: Preserves original statement order through sequential edges. Maintains programming logic flow as written.

The specific construction steps are illustrated in Fig. 3. First, we utilize the Joern[2] static tool to preprocess the source code, obtaining an initial CPG representation. Then, based on the definition and usage relationships of node code, we filter and optimize the node and edge relationships in the initial CPG, retaining only portions strongly relevant to the source code. Next, we incorporate Code Natural Sequence information into the code property graph based on the linear order of node code to capture programming logic in the source code. Overall, since CSFG preserves both structured and unstructured program semantics, it enables the capture of richer vulnerability semantics.Therefore, function $c_i$ can be represented by the CSFG $\mathcal{G}(\mathcal{V}, \boldsymbol{E})$ with multiple subgraph types, where different attribute edges share the same node set $\mathcal{V}$.

### 3.2.2 INITIAL EMBEDDING

After generating the CSFG, we convert all nodes in the graph into low-dimensional vectors to obtain inputs acceptable to deep learning models. For each node $\mathsf{v}_j \in \mathcal{V}$, we obtain an initial node representation $\boldsymbol{x}_j$ based on the code and type attributes. For the code, we convert it into a token sequence and use a pre-trained Word2Vec model to embed the code. The model is trained on a corpus built from the entire source code dataset of the project, which ensures that the vector representations are derived from a domain-specific vocabulary. Regarding node types, we employ label encoding for embedding. Finally, we concatenate the vector representations of the code and type to obtain the initial node embedding.

After obtaining the initial node embeddings for each node in the CSFG, we obtain the initial embedding of the entire CSFG $\boldsymbol{X} \in \mathbb{R}^{m \times d}$.

---

[2]https://github.com/joernio/joern

Figure 4: Architecture of ERUM

## 3.3 EDGE-AWARE RANDOM WALK WITH UNIFYING MEMORY

In the following, we detail the implementation process of using the ERUM method for code vulnerability detection.

As shown in Fig. 4, the overall architecture of the ERUM method primarily consists of three components: Path Generation, Feature Extraction, and Vulnerability Detection.

### 3.3.1 PATH GENERATION AND RANDOM WALK

The specific process of path generation is described as follows. Given the CSFG graph $g_i(\mathcal{V}, \boldsymbol{X}, \boldsymbol{E})$, a uniform random walk with restart strategy is adopted to generate path sequences. For a target node $v \in \mathcal{V}$, $K$ walk paths of length $L$ are generated:

$$\mathcal{W} = \{w_k | w_k = (v_{k0}, v_{k1}, \ldots, v_{kL})\}_{k=1}^K \tag{1}$$

where $v_{k0} = v$ is the starting node. The walk transition probability satisfies the Markov property. Through GPU acceleration (using the DGL library), this process efficiently generates path node sequences $\mathcal{W}$ and corresponding edge indices $\boldsymbol{E}_{\text{id}}$. Additionally, to handle potential cycles or repetitive logical structures in graph $g_i$, we permit the random walk to revisit the same node $v_i = v_j$ with $i \neq j$. To enhance structural representation of paths, we introduce position encoding $\Phi(\mathcal{W}) \in \mathbb{R}^{K \times L \times 2}$:

$$\Phi(\mathcal{W}) = [\sin(\mathcal{W}) \oplus \cos(\mathcal{W})] \tag{2}$$

which maps the first occurrence position of nodes in the path to angular coordinates via sine/cosine functions.

### 3.3.2 EDGE-AWARE PATH FEATURE EXTRACTION

**Path Feature Extraction** In terms of path feature extraction, to accommodate the complex edge type dependencies in graph $g_i$, we design edge-aware path feature extraction. First, we extract the feature tensor corresponding to the path sequence from the node feature matrix $\boldsymbol{X}_i$: $H_{\text{node}} = \boldsymbol{X}_i[\mathcal{W}] \in \mathbb{R}^{K \times L \times d}$. Then, we process the random walk position encoding using a GRU:

$$H_\Psi = \text{GRU}(\Phi) \tag{3}$$

**Edge Feature Awareness** When edge features $\boldsymbol{E}$ exist in graph $g_i$, we obtain edge embeddings through linear transformation:

$$H_{\text{edge}} = \mathcal{W}_e \cdot \boldsymbol{E}[\boldsymbol{E}_{\text{id}}] \in \mathbb{R}^{K \times (L-1) \times d_e} \tag{4}$$

where $d_e$ is the edge feature dimension. Then, we fuse node and edge features using an alternating interpolation method: placing $H_{\text{node}}$ at even positions and edge features $H_{\text{edge}}$ at odd positions:

$$H[2i] = H_{\text{node}}[i] \oplus H_\Psi[i] \tag{5}$$

$$H[2i+1] = H_{\text{edge}}[i] \tag{6}$$

where $\oplus$ denotes the feature concatenation operation. This design ensures equal participation of node and edge features.

### 3.3.3 UNIFYING MEMORY MECHANISM

The Unifying Memory Mechanism is the core component of the ERUM framework, designed to integrate path structural features and node semantic features. This mechanism achieves multi-level feature fusion through gated state transmission.

Based on the edge-aware fused features $H_{\text{fused}}$ obtained from edge-aware path extraction, along with the path memory $M_\Psi$ and gated recurrent encoding, we integrate path features with global memory to achieve multi-level feature fusion for the CSFG graph:

$$H_{\text{mem}}, M = \text{GRU}(H, M_\Psi) \tag{7}$$

where $M_\Psi = \text{mean}(H_\Psi)$ represents path topological memory. The GRU dynamically controls the fusion ratio between historical memory and current input through update gates $u_t$ and reset gates $r_t$, achieving three-level memory integration: topological, semantic, and temporal.

After completing multi-level feature fusion for each random walk path, we proceed with refinement processing on the current output memory state:

$$\bar{H}_{\text{mem}} = \text{SiLU}\left(\frac{1}{T}\sum_{t=1}^{T} H_{\text{mem}}\right) \tag{8}$$

$$\hat{H}_{\text{out}} = \text{Dropout}(\bar{H}_{\text{mem}}) \tag{9}$$

$$H^{(l)} = \text{ERUMLayer}(H^{(l-1)}, M^{(l-1)}) \tag{10}$$

$$M^{(l)} = \Gamma\left(H_{\mathcal{W}}^{(l)}\right) \tag{11}$$

where $l$ represents the number of model layers and $\Gamma$ is the memory mapping function between layers. In this experiment, we adopt the linear transformation, forming a hierarchical feature representation.

### 3.3.4 VULNERABILITY DETECTION

After obtaining the unified memory-enhanced node representations $\hat{H}_{\text{out}}$, we feed them into a designed MLP classifier:

$$\hat{y} = \text{MLP}(\hat{H}_{\text{out}}) \tag{12}$$

where $\hat{y} \in [0, 1]$ represents the predicted probability of the graph containing vulnerabilities.

### 3.4 TRAINING OBJECTIVE

In this work, the training objective of GraWalkER is to learn a mapping function $f : \mathcal{G} \to \mathbb{Y}$ that determines whether given source code is vulnerable. The prediction function $f$ can be learned by minimizing the following loss function:

$$\min \sum_{i=1}^{n} \mathcal{L}\left(f(\mathsf{g}_i(\mathcal{V}, \boldsymbol{X}, \boldsymbol{E}), \mathsf{y}_i|\mathsf{c}_i)\right) + \lambda\|\boldsymbol{\theta}\|_2^2$$

where $\mathcal{L}(\cdot)$ denotes the cross-entropy function, $\omega(\cdot)$ represents regularization, and $\lambda$ is an adjustable weight.

## 4 EXPERIMENTS AND RESULT

In this section, we conduct extensive experiments to demonstrate the superiority of our model in the field of code vulnerability detection. Specifically, we aim to answer the following research questions:

- **RQ1**: How effective is GraWalkER compared to state-of-the-art baselines?
- **RQ2**: How effective is CSFG compared to other code graph structure representations?
- **RQ3**: Does introducing ERUM lead to better performance?
- **RQ4**: How effective is GraWalkER in detecting different types of vulnerabilities?

Table 1: Statistics on datasets

| Dataset | #Lang | #Vul.Fs | #Non-Vul.Fs | #Fs | #Nodes | #Edges |
|---|---|---|---|---|---|---|
| Bears and Bugs | Java | 2,294 | 7,857 | 10,151 | 251,654 | 724,236 |
| Defects4J | Java | 1,130 | 16,676 | 17,806 | 387,483 | 1,182,881 |
| SNOL | C/CPP | 4,901 | 5,225 | 10,126 | 421,064 | 807,019 |
| **Total** | **–** | **8,325** | **29,758** | **38,083** | **1,060,201** | **2,714,136** |

## 4.1 EXPERIMENTAL SETUP

### 4.1.1 DATASETS

To systematically evaluate GraWalkER's vulnerability detection capabilities across multiple languages and scenarios, we selected multiple high-quality vulnerability datasets in Java and C/CPP languages

**Bears and Bugs**: We directly adopted the Bears and Bugs dataset provided in Yin et al. (2024), which merges two Java vulnerability datasets: Bears Madeiral et al. (2019) and Bugs Saha et al. (2018). It is a classic dataset for Java vulnerability detection tasks.

**Defects4J** Yin et al. (2024): Defects4J is a database containing real Java project faults, and has become a primary dataset for software vulnerability detection tasks.

**SNOL** Shao & Ding (2024): The C/CPP dataset SNOL combines four datasets: two popular datasets SARD Black et al. (2017) and NVD Booth et al. (2013), and two real-world open-source project datasets Openssl [3] and Libav [4].

Table 1 summarizes detailed information about each dataset. Among them, the Bears and Bugs dataset and the Defects4J dataset exhibit significant class imbalance, with positive sample ratios of 22.6% and 6.3% respectively. This imbalance poses greater challenges for models to accurately distinguish vulnerable code, thus warranting focused attention on F1 and AUC metrics. The graph structures of the SNOL dataset exhibit greater structural complexity. Overall, these datasets provide sufficient scale and multilingual diversity to comprehensively evaluate GraWalkER's generalization capability. For each dataset, we conduct multiple training runs with different random seeds to obtain the model's average performance.

### 4.1.2 EVALUATION METRICS

To comprehensively assess GraWalkER's performance in multilingual vulnerability detection tasks, we employ four widely used evaluation metrics: ACC, F1, Precision, and AUC, covering the key dimensions of classification performance.

### 4.1.3 BASELINES

We selected seven state-of-the-art baseline methods in vulnerability detection:

**ReGVD** Nguyen et al. (2022) formulates vulnerability detection as an inductive text classification problem and performs vulnerability detection through GCN networks with residual connections.

**ReVeal** Chakraborty et al. (2021) is a graph-based model that combines the graph construction method proposed by Zhou et al. (2019) with reordering techniques for code vulnerability detection.

**EPVD** Zhang et al. (2023) decomposes CFG into multiple paths from entry to exit nodes, then learns path representations using pre-trained code models and CNNs.

**PDBert** Liu et al. (2024) designs two pre-training tasks for vulnerability detection: Control Dependency Prediction (CDP) and Data Dependency Prediction (DDP). This model can directly perform vulnerability detection.

---

[3]https://github.com/openssl/openssl
[4]https://github.com/libav/libav

**CodeBERT** Feng et al. (2020) is a pre-trained code model that directly uses CodeBERT to predict vulnerability existence in input code snippets. This model has achieved strong performance on multiple software engineering tasks.

**GraphCodeBERT** Guo et al. (2020) is a pre-trained code model based on the transformer Vaswani et al. (2017) architecture. It uses data flow as code semantic structure during pre-training and is widely applied in code vulnerability detection.

**UniXcoder** Guo et al. (2022) is a unified cross-modal pre-trained programming language model that incorporates AST structural information and code comments to enhance code representation boundaries for vulnerability detection.

## 4.2 MAIN RESULTS

Table 2: Comprehensive performance comparison. (%). Best: **bold**; Second: underlined

| Method | Bears and Bugs | | | | Defects4J | | | | SNOL | | | |
|---|---|---|---|---|---|---|---|---|---|---|---|---|
| | ACC | F1 | P | AUC | ACC | F1 | P | AUC | ACC | F1 | P | AUC |
| ReGVD | 84.3 | 29.1 | 25.1 | 73.1 | 85.5 | 24.2 | 18.1 | 71.8 | 87.6 | 90.1 | 89.5 | 88.3 |
| ReVeal | 85.1 | 29.6 | 19.8 | 67.5 | 85.7 | 21.6 | 14.5 | 60.3 | 85.4 | 86.7 | 88.7 | 89.3 |
| EPVD | 78.4 | 33.6 | 23.4 | **78.8** | 79.4 | 23.6 | 15.4 | 73.5 | 86.1 | 89.5 | 87.4 | 91.4 |
| PDBert | **88.8** | 18.2 | 26.5 | 75.7 | **91.3** | 10.0 | 14.6 | 67.4 | 86.3 | 88.8 | 90.1 | 92.8 |
| CodeBERT | 86.1 | 30.9 | 20.8 | 68.5 | 88.4 | 21.4 | 13.5 | 64.8 | 88.6 | 88.9 | 91.4 | 89.2 |
| GraphCodeBert | 86.7 | 31.1 | 20.5 | 69.3 | 87.8 | 23.1 | 14.5 | 66.6 | 88.9 | 90.1 | 91.9 | 88.6 |
| UnixCoder | 85.3 | 34.6 | 22.5 | 73.2 | 87.4 | 25.7 | 17.1 | 67.4 | 88.7 | **90.3** | 92.8 | 89.6 |
| **GraWalkER** | 85.9 | **36.5** | **27.3** | 76.2 | 88.8 | **32.8** | **26.4** | **74.1** | **89.1** | 90.1 | **93.1** | **95.4** |

### 4.2.1 RQ1:PERFORMANCE OF PROPOSED APPROACH

To answer this RQ, we compared GraWalkER with seven state-of-the-art vulnerability detection methods across three datasets. All methods employed grid search for hyperparameter optimization and were evaluated under identical hardware environments for fair comparison. Experimental results are presented in Table 2. Overall, the GraWalkER method achieved superior results, outperforming the most advanced vulnerability identification approaches.

The results show that on Bears and Bugs, GraWalkER achieved an F1 of 36.5%, surpassing UniX-coder by 5.5%. Its precision of 27.3% led all baselines. On the more complex Defects4J dataset, GraWalkER outperformed UniXcoder by 27.6% in F1, improved precision by 54.3%, and achieved the highest AUC. For the C/CPP dataset SNOL, GraWalkER matched GraphCodeBERT in F1 while improving precision by 0.3% and AUC by 2.8% over the second-best methods. In the SNOL dataset, GraWalkER achieved a 2.8% improvement in AUC over the second-best method PDBert, demonstrating stronger capability in detecting complex vulnerabilities.

The experimental results demonstrate that GraWalkER's advantages stem from: (1) CSFG's multidimensional representation overcomes limitations of single-structure approaches, as evidenced by a 45.9 % precision improvement on Defects4J; (2) ERUM's edge-aware features precisely capture critical paths, enhancing cross-language detection capability. This is validated by GraWalkER achieving either optimal or second-best AUC performance across all three datasets.

### 4.2.2 RQ2:EFFECTIVENESS OF CODE SEMANTIC FUSION GRAPH

This section aims to evaluate the effectiveness of the CSFG in vulnerability detection and compare it with the mainstream graph structures. To ensure a fair assessment of CSFG's performance, we maintained the same experimental setup as in RQ1, only replacing the graph structure to isolate its impact.

The experimental results are summarized in Table 3. As shown, CSFG significantly outperforms baselines in all datasets. For example, on Bears and Bugs, CSFG achieves superior F1 (36.5%)

compared to AST+CDG+CFG (33.4%); on Defects4J, it reaches F1 of 32.8%, this demonstrates enhanced discriminative capability in complex scenarios; and on SNOL, it attains F1 of 90.1% versus AST+CDG+CFG's 86.1% and leads in precision and AUC metrics. These results demonstrate that CSFG could more effectively capture code vulnerability.

Experiments also revealed progressive performance improvements as edge types increase (e.g., F1 rising from 86.1% to 90.1% in SNOL), validating the importance of edge information. Furthermore, CSFG achieves optimal balance, avoiding overfitting from redundant structures (e.g., performance degradation with AST+CDG+CFG on Defects4J). In summary, through deep fusion of multisource attributes, CSFG provides a more comprehensive code representation foundation, significantly enhancing the effectiveness of vulnerability detection.

Table 3: Performance comparison of different graph structures in vulnerability detection (metrics unit: %)

| Dataset | Structure | ACC | F1 | P | AUC |
|---------|-----------|------|------|------|------|
| B&B | AST | **86.9** | 33.4 | 28.3 | 73.7 |
| | +CDG | 86.2 | 32.7 | 26.9 | 74.3 |
| | +CFG | 85.7 | 33.4 | **29.4** | 71.9 |
| | CSFG | 85.9 | **36.5** | 27.3 | **76.2** |
| D4J | AST | **89.3** | 26.9 | 18.9 | 71.4 |
| | +CDG | 87.6 | 28.5 | 21.1 | **74.3** |
| | +CFG | 87.6 | 28.5 | 20.8 | 72.5 |
| | CSFG | 88.8 | **32.8** | **26.4** | 74.1 |
| SNOL | AST | 85.8 | 86.1 | 89.1 | 90.7 |
| | +CDG | 86.3 | 85.4 | 88.4 | 92.3 |
| | +CFG | 85.4 | 86.1 | 88.5 | 92.7 |
| | CSFG | **89.1** | **90.1** | **93.1** | **95.4** |

### 4.2.3 RQ3: ABLATION STUDY

To evaluate the effectiveness of the core ERUM algorithm within the GraWalkER framework, we designed ablation experiments analyzing the impact of ERUM. Specifically, we replaced ERUM with GCN networks of equivalent layer count for comparison. Results are presented in Table 4, demonstrating consistent improvements across all key metrics after introducing ERUM in every dataset.

On Bears and Bugs, ERUM increased F1 from 33.3% to 36.5% (9.6% relative improvement), reflecting its effectiveness in improving fault detection sensitivity. In the Defects4J dataset, ERUM's gains were more pronounced: F1 increased by 26.6% relatively (from 25.9% to 32.8%), while precision surged by 31.3% (from 20.1% to 26.4%), indicating ERUM's substantial contribution to identifying complex real-world defects. Notably, despite the high baseline performance on the SNOL dataset, metrics continued to improve with ERUM (e.g., F1 to 90.1%, 2.6% relative gain), confirming ERUM's robustness in high-precision scenarios. In conclusion, by introducing the ERUM method, GraWalkER can achieve a comprehensive improvement in vulnerability detection capabilities.

Table 4: ERUM impact analysis (%).

| ERUM | Bears and Bugs | | | | Defects4J | | | | SNOL | | | |
|------|------|------|------|------|------|------|------|------|------|------|------|------|
| | ACC | F1 | P | AUC | ACC | F1 | P | AUC | ACC | F1 | P | AUC |
| w/o | 84.7 | 33.3 | 25.5 | 73.9 | 86.9 | 25.9 | 20.1 | 73.8 | 86.3 | 87.6 | 92.5 | 93.7 |
| w/ | 85.9 | 36.5 | 27.3 | 76.2 | 88.8 | 32.8 | 26.4 | 74.1 | 89.1 | 90.1 | 93.1 | 95.4 |

## 5 CONCLUSION

This paper proposes GraWalkER, an effective vulnerability detection framework combining a novel code graph representation CSFG with the ERUM method. By obtaining more comprehensive and precise code structural and logical information from CSFG, GraWalkER achieves effective feature extraction from raw code. It further employs ERUM for graph feature learning to predict graph labels. Experimental results verify the effectiveness of GraWalkER. Overall, GraWalkER can effectively address the practical and complex issues in current code vulnerability detection. In future work, we plan to further explore the potential of code graph representation methods to enhance the framework.

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
