# OpenReview forum: "GraWalkER: Vulnerability Detection via Code Semantic Fusion Graph with Edge-Aware Random Walk"
_ICLR.cc/2026/Conference — Submitted to ICLR 2026_

### Official Review · Reviewer_DiNN · 2025-10-24

**Soundness:** 3
**Presentation:** 3
**Contribution:** 2
**Rating:** 4
**Confidence:** 3

**Summary:**

The author proposes a method for generating fusion graph features in this paper. Furthermore, a new ERUM scheme is introduced to achieve feature extraction for graph paths. The authors conduct extensive experiments on three datasets across Java and C/C++ code, benchmarking against seven state-of-the-art methods. Results show that GraWalkER consistently outperforms baselines in F1, AUC, and precision. Ablation studies also support this method’s efficacy.

**Strengths:**

This approach is technically sound and builds upon relevant prior work. The paper effectively highlights the weaknesses of standard GNN models on heterogeneous code graphs. The proposed solution addresses these issues to a certain extent.

**Weaknesses:**

The practical significance of the improvements in this proposal remains to be weighed. In absolute terms, some improvements are modest. For instance, on the SNOL dataset, GraWalkER's F1 score (90.1%) remains largely on par with the best baseline (90.3%), with gains primarily observed in precision and AUC. On Defects4J, despite a relatively larger relative improvement, the final F1 score remains only around 33%, indicating that numerous defects continue to be missed.

It would be beneficial to include some discussion or experimentation regarding computational costs. How does GraWalkER compare to typical GNNs or CodeBERT-based approaches in terms of training time or model size? If the random walk method incurs higher computational costs, is the improvement in accuracy justified?

This paper compares its own data fusion approach with three alternative methods. However, the comparative results suggest that this novel data fusion scheme does not appear to yield any significant advantages. It may be worth considering, in subsequent experiments, incorporating performance comparisons of different combinations of graph features to determine which features contribute most significantly to the recognition task.

**Questions:**

see above.
(This paper was previously submitted to AAAI, but it appears only the formatting was revised.)

---

### Official Review · Reviewer_D8CL · 2025-10-29

**Soundness:** 3
**Presentation:** 3
**Contribution:** 2
**Rating:** 4
**Confidence:** 3

**Summary:**

This paper proposes GraWalkER, a framework for code vulnerability detection that integrates a Code Semantic Fusion Graph (CSFG) and an Edge-aware Random Walk with Unifying Memory (ERUM) method. The CSFG combines multiple code representations—AST, CFG, CDG, Call Graph, and Code Natural Sequence—to capture both semantic and structural information. The ERUM component replaces traditional GNNs by performing edge-type-aware random walks to extract path features, enabling comprehensive graph representation learning. The model is evaluated on three datasets (Bears and Bugs, Defects4J, SNOL) covering Java and C/CPP languages. Experimental results demonstrate that GraWalkER outperforms several state-of-the-art baseline methods, demonstrating effectiveness in detecting vulnerabilities in complex code scenarios.

**Strengths:**

1. This paper has a clear problem definition of vulnerability detection for code.
2. The proposed CSFG aims to integrate semantic and structural information, which are both important to vulnerability detection for code.
3. The experiments are comprehensive to show the improvement of GraWalkER.

**Weaknesses:**

1. The motivation of this paper is unclear. Although this paper mentions that existing methods fail to correctly interpret semantics, it does not provide the corresponding reasons and detailed analysis. Moreover, the example of Fig.1 is not fully explained, which is confusing.
2. The introduction introduces the issue of over-smoothing, which is a widely studied topic. What are the advantages of the ERUM proposed in this article compared to existing solutions?
3. What is the complexity of constructing various graphs in this paper? The author should provide a detailed analysis.
4. In Table2, GraWalkER does not surpass existing SOTA in some metrics, and the author does not seem to explain why this situation occurs.
5. It would be better if the author could provide a visualization to illustrate how GraWalkER captures semantic information.

**Questions:**

Please see weakness.

---

### Official Review · Reviewer_nBgr · 2025-10-31

**Soundness:** 2
**Presentation:** 3
**Contribution:** 2
**Rating:** 4
**Confidence:** 4

**Summary:**

The paper presents GraWalkER, a two-stage graph-based framework for function-level vulnerability detection. In the first stage, a Code Semantic Fusion Graph (CSFG) is constructed by integrating AST, CFG, CDG, CG, and CNS. In the second stage, an Edge-aware Random Walk with Unifying Memory (ERUM) mechanism is proposed to extract path features for graph classification. The authors evaluate GraWalkER on three datasets (Java and C/C++) and compare it against seven baselines, claiming state-of-the-art performance.

**Strengths:**

**Robust Code Representation Integration**: The CSFG is a technically solid method for feature engineering, effectively addressing the "incomplete code representation" problem by integrating five complementary views of the code, including the non-structural Code Natural Sequence.

**Strong Ablation & Empirical Results**: The ablation studies clearly validate the contribution of both the CSFG's multi-modal representation and the ERUM module. The performance gains reported, particularly on the complex Defects4J dataset, suggest that the proposed architecture is highly effective in practice.

**Weaknesses:**

**Limited Conceptual Novelty in Graph Fusion**: The central mechanism of fusing multiple code graphs (AST, CFG, CDG) into a single, heterogeneous CPG structure is an established technique in the field. The novelty of CSFG is largely incremental, consisting of adding the Call Graph (CG) and Code Natural Sequence (CNS) edges to a typical CPG base. This conceptual approach is highly similar to existing multi-modal/multi-graph fusion methods, such as MVulD [1].

**Modern PLM/LLM Baselines are Missing**: While the paper includes older PLMs (CodeBERT, UniXcoder), it fails to compare against the latest PLM-based [1-3] and LLM-based [4] implementations specifically fine-tuned for this task, which represent the current empirical SOTA.

[1] Abundant Modalities Offer More Nutrients: Multi-Modal-Based Function-level Vulnerability Detection

[2] Distinguishing Look-Alike Innocent and Vulnerable Code by Subtle Semantic Representation Learning and Explanation

[3] Linevul: A transformer-based line-level vulnerability prediction

[4] Boosting Vulnerability Detection of LLMs via Curriculum Preference Optimization with Synthetic Reasoning Data

**Questions:**

1. Can the authors precisely articulate the conceptual novelty of CSFG compared to existing state-of-the-art CPG/multi-graph fusion models? What fundamental theoretical limitation of prior multi-graph models does the CSFG framework resolve.

2. Required Baselines for SOTA Claim: The absence of key modern baselines is a major weakness. Please include experimental results comparing GraWalkER against the highly competitive methods.

---

### Official Review · Reviewer_oGJK · 2025-11-01

**Soundness:** 2
**Presentation:** 3
**Contribution:** 2
**Rating:** 4
**Confidence:** 4

**Summary:**

This paper presents GraWalkER, a novel framework for software vulnerability detection that integrates a Code Semantic Fusion Graph with an Edge-aware Random Walk with Unifying Memory (ERUM).
Experiments on three large datasets show that GraWalkER consistently outperforms popular baselines, demonstrating improved precision and generalization across Java and C/C++ code.

**Strengths:**

1. The presentation of this paper is clear, and easy to read and understand.
2. The combination of a multi-source code representation and a random-walk-based learning paradigm is creative and effectively addresses the limitations of conventional GNNs in modeling code structure and semantics.
3. The paper conducts systematic experiments on multiple datasets and programming languages, comparing against diverse baselines and providing clear evidence of the method’s effectiveness.

**Weaknesses:**

1. This paper lacks of comparison with some LLM-based approaches. Specifically, recent LLMs, such as Qwen series, Claude series, and GPT series models, and some LLM-based vulnerability detection approaches, such as GRACE [1] have shown remarkable performance on code understanding and vulnerability detection tasks. The paper only compares with medium-scale pretrained models, but not with current LLM-based methods, which limits the claim of SOTA performance.
2. Another concern is about the complexity and scalability of the proposed method. The combination of CSFG and ERUM may be computationally heavy for large projects or real-time analysis.
The paper would benefit from runtime or scalability analysis to justify practical deployability.
3. This paper is also missing some details on hyperparameters and architecture tuning. Specifically, this paper does not specify details. This limits the reproducibility and interpretability of the results.
4. The ablation isolates CSFG and ERUM, but not finer components (e.g., impact of individual edge types). Additional experiments would better clarify what parts contribute most.

**Reference**

[1] Guilong Lu, Xiaolin Ju, Xiang Chen, Wenlong Pei, and Zhilong Cai. 2024. GRACE: Empowering LLM-based software vulnerability detection with graph structure and in-context learning. Journal of Systems and Software 212 (2024), 112031.

**Questions:**

Please refer to the concerns in Weaknesses.

---

### Meta-Review · Area_Chair_39P2 · 2026-01-07

**Summary:**

The reviewers are concerned about the weak motivation, lack of novelty, important baseline missing, and weak experimental results.

**Reviewer Concerns:**

There is no rebuttal provided.

**Reviewer Scores:**

The scores will be the same as there is no rebuttal provided.

---

### Decision · Program_Chairs · 2026-01-26

Reject